# Hessian-based Analysis of Large Batch Training and Robustness to Adversaries

**Zhewei Yao**[1][*] **Amir Gholami**[1][*] **Qi Lei**[2] **Kurt Keutzer**[1] **Michael W. Mahoney**[1]

[1] University of California at Berkeley, {zheweiy, amirgh, keutzer and mahoneymw}@berkeley.edu
[2] University of Texas at Austin, leiqi@ices.utexas.edu

## Abstract

Large batch size training of Neural Networks has been shown to incur accuracy loss when trained with the current methods. The exact underlying reasons for this are still not completely understood. Here, we study large batch size training through the lens of the Hessian operator and robust optimization. In particular, we perform a Hessian based study to analyze exactly how the landscape of the loss function changes when training with large batch size. We compute the true Hessian spectrum, without approximation, by back-propagating the second derivative. Extensive experiments on multiple networks show that saddle-points are not the cause for generalization gap of large batch size training, and the results consistently show that large batch converges to points with noticeably higher Hessian spectrum. Furthermore, we show that robust training allows one to favors flat areas, as points with large Hessian spectrum show poor robustness to adversarial perturbation. We further study this relationship, and provide empirical and theoretical proof that the inner loop for robust training is a saddle-free optimization problem. We present detailed experiments with five different network architectures, including a residual network, tested on MNIST, CIFAR-10, and CIFAR-100 datasets.

## 1 Introduction

During the training of a Neural Network (NN), we are given a set of input data $\mathbf{x}$ with the corresponding labels $y$ drawn from an unknown distribution $\mathcal{P}$. In practice, we only observe a set of discrete examples drawn from $\mathcal{P}$, and train the NN to learn this unknown distribution. This is typically a non-convex optimization problem, in which the choice of hyper-parameters would highly affect the convergence properties. In particular, it has been observed that using large batch size for training often results in convergence to points with poor convergence properties. The main motivation for using large batch is the increased opportunities for data parallelism which can be used to reduce training time [13]. Recently, there have been several works that have proposed different methods to avoid the performance loss with large batch [16, 28, 31]. However, these methods do not work for all networks and datasets. This has motivated us to revisit the original problem and study how the optimization with large batch size affects the convergence behavior.

We first start by analyzing how the Hessian spectrum and gradient change during training for small batch and compare it to large batch size and then draw connection with robust training. In particular, we aim to answer the following questions:

**Q1** How is the training for large batch size different than small batch size? Equivalently, what is the difference between the local geometry of the neighborhood that the model converges when large batch size is used as compared to small batch?

**A1** We backpropagate the second-derivative and compute its spectrum during training. The results show that despite the arguments regarding prevalence of saddle-points plaguing optimization [6, 12], that is actually not the problem with large batch size training, even when batch size is increased to the gradient descent limit. In [19], an approximate numerical method was used to approximate the

---

[*]Equal contribution

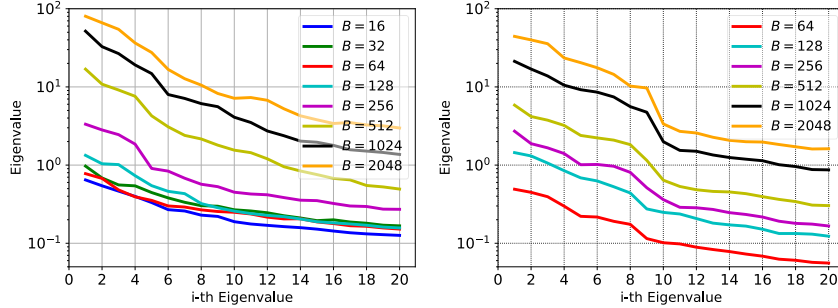

**Figure 1:** *Top 20 eigenvalues of the Hessian is shown for C1 on CIFAR-10 (left) and M1 on MNIST (right) datasets. The spectrum is computed using power iteration with relative error of* 1E-4.

maximum at a point. Here, by directly computing the spectrum of the true Hessian, we show that large batch size progressively gets trapped in areas with noticeably larger spectrum (and not just the dominant eigenvalue). For details please see §2, especially Figs. 1, 2 and 4.

**Q2** What is the connection between robust optimization and large batch size training? Equivalently, how does the batch size affect the robustness of the model to adversarial perturbation?

**A2** We show that robust optimization is antithetical to large batch training, in the sense that it favors areas with small spectrum (aka flat minimas). We show that points converged with large batch size are significantly more prone to adversarial attacks as compared to a model trained with small batch size. Furthermore, we show that robust training progressively favors the opposite, leading to points with flat spectrum and robust to adversarial perturbation. We provide empirical and theoretical proof that the inner loop of the robust optimization, where we find the worst case, is a saddle-free optimization problem. Details are discussed in §3, especially Table 1, 7 and Figs. 4, 6.

**Limitations:** We believe it is critical for every paper to clearly state limitations. In this work, we have made an effort to avoid reporting just the best results, and repeated all the experiments at least three times and found all the findings to be consistent. Furthermore, we performed the tests on multiple datasets and multiple models, including a residual network, to avoid getting results that may be specific to a particular test. The main limitation is that we do not propose a solution for large batch training. Even though we show a very promising connection between large batch and robust training, but we emphasize that this is an analysis paper to understand the original problem. There has been several solutions proposed so far, but they only work for particular cases and require extensive hyper-parameter tuning. We are performing an in-depth follow up study to use the results of this paper to guide large batch size training.

**Related Work.** Deep neural networks have achieved good performance for a wide range of applications. The diversity of the different problems that a DNN can be used for, has been related to their efficiency in function approximation [25, 7, 21, 1]. However the work of [32] showed that not only the network can perform well on real dataset, but it can also memorize randomly labeled data very well. Moreover, the performance of the network is highly dependent on the hyper-parameters used for training. In particular, recent studies have shown that Neural Networks can easily be fooled by imperceptible perturbation to input data [15]. Moreover, multiple studies have found that large batch size training suffers from poor generalization capability [16, 31].

Here we focus on the latter two aspects of training neural networks. [19] presented results showing that large batches converge to a "sharper minima". It was argued that even if the sharp minima has the same training loss as the flat one, but small discrepancies between the test data and the training data can easily lead to poor generalization performance [19, 9]. The fact that "flat minimas" generalize well goes back to the earlier work of [18]. The authors relate flat minima to the theory of minimum description length [26], and proposed an optimization method to actually favor flat minimas. There have been several similar attempts to change the optimization algorithm to find "better" regions [8, 5]. For instance, [5] proposed entropy-SGD, which uses Langevin dynamics to augment the loss functional to favor flat regions of the "energy landscape". The notion of flat/sharpness does not have a precise definition. A detailed comparison of different metrics is discussed in [9], where the authors show that sharp minimas can also generalize well. The authors also argued that the sharpness can be arbitrarily changed by reparametrization of the weights. However, this won't happen when

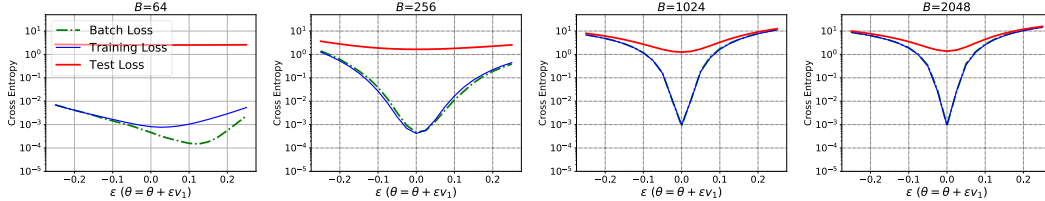

**Figure 2:** *The landscape of the loss is shown along the dominant eigenvector, $v_1$, of the Hessian for C1 on CIFAR-10 dataset. Here $\epsilon$ is a scalar that perturbs the model parameters along $v_1$.*

considering the same model and just changing the training hyper-parameters which is the case here. In [28, 29], the authors proposed that the training can be viewed as a stochastic differential equation, and argued that the optimum batch size is proportional to the training size and the learning rate.

As our results show, there is an interleaved connection by studying when NNs do not work well. [30, 15] found that they can easily fool a NN with very good generalization by slightly perturbing the inputs. The perturbation magnitude is most of the time imperceptible to human eye, but can completely change the networks prediction. They introduced an effective adversarial attack algorithm known as Fast Gradient Sign Method (FGSM). They related the vulnerability of the Neural Network to linear classifiers and showed that RBF models, despite achieving much smaller generalization performance, are considerably more robust to FGSM attacks. The FGSM method was then extended in [20] to an iterative FGSM, which performs multiple gradient ascend steps to compute the adversarial perturbation. Adversarial attack based on iterative FGSM was found to be stronger than the original one step FGSM. Various defenses have been proposed to resist adversarial attacks [24, 14, 17, 2, 11]. We will later show that there is an interleaved connection between robustness of the model and the large batch size problem.

The structure of this paper is as follows: We present the results by first analyzing how the spectrum changes during training, and test the generalization performances of the model for different batch sizes in §2. In section §3, we discuss details of how adversarial attack/training is performed. In particular, we provide theoretical proof that finding adversarial perturbation is a saddle-free problem under certain conditions, and test the robustness of the model for different batch sizes. Also, we present results showing how robust training affects the spectrum with empirical studies. Finally, in section §4 we provide concluding remarks.

## 2   Large Batch, Generalization Gap and Hessian Spectrum

**Setup:** The architecture for the networks used is reported in Table 6. In the text, we refer to each architecture by the abbreviation used in this table. Unless otherwise specified, each of the batch sizes are trained until a training loss of 0.001 or better is achieved. Different batches are trained under the same conditions, and no weight decay or dropout is used.

We first focus on large batch size training versus small batch and report the results for large batch training for C1 network on CIFAR-10 dataset, and M1 network on MNIST are shown in Table 1, and Table 7, respectively. As one can see, after a certain point increasing batch size results in performance degradation on the test dataset. This is in line with results in the literature [19, 16].

As discussed before, one popular argument about large batch size's poor generalization accuracy has been that large batches tend to get attracted to "sharp" minimas of the training loss. In [19] an approximate metric was used to measure curvature of the loss function for a given model parameter. Here, we directly compute the Hessian spectrum. Note that computing the whole Hessian matrix is infeasible as it is a $\mathcal{O}(N^2)$ matrix. However, the spectrum can be computed using power iteration by back-propagating the matvec of the Hessian [23]. Unless otherwise noted, we continue the power iterations until a relative error of 1E-4 reached for each individual eigenvalue.

With this approach, we have computed the first top 20 eigenvalues of the Hessian for different batch sizes as shown in Fig. 1. Moreover, the value of the dominant eigenvalue, denoted by $\lambda_1^\theta$, is reported in Table 1, and Table 2, respectively (Additional result for MNIST tested using LeNet-5 is given in appendix. Please see Table 7). From Fig. 1, we can clearly see that for all the experiments, large batches have a noticeably larger Hessian spectrum both in the dominant eigenvalue as well as the rest of the 19 eigenvalues. However, note that curvature is a very local measure. It would be more

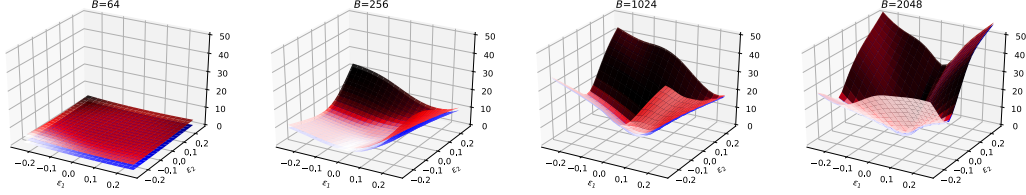

**Figure 3:** *The landscape of the loss is shown when the C1 model parameters are changed along the first two dominant eigenvectors of the Hessian with the perturbation magnitude $\epsilon_1$ and $\epsilon_2$.*

informative to study how the loss functional behaves in a neighborhood around the point that the model has converged. To visually demonstrate this, we have plotted how the total loss changes when the model parameters are perturbed along the dominant eigenvector as shown in Fig. 2, and Fig. 7 for C1 and M1 models, respectively. We can clearly see that the large batch size models have been attracted to areas with higher curvature for both the test and training losses.

This is reflected in the visual figures. We have also added a 3D plot, where we perturb the parameters of C1 model along both the first and second eigenvectors as shown in Fig. 3. The visual results are in line with the numbers shown for the Hessian spectrum (see $\lambda_1^\theta$) in Table 1, and Table 7. For instance, note the value of $\lambda_1^\theta$ for the training and test loss for $B = 512,\ 2048$ in Table 1 and compare the corresponding results in Fig. 3.

A recent argument has been that saddle-points in high dimension plague optimization for neural networks [6, 12]. We have computed the dominant eigenvalue of the Hessian along with the total gradient during training and report it in Fig. 4. As we can see, large batch size progressively gets attracted to areas with larger spectrum, but it clearly does not get stuck in saddle points since the gradient is still large.

**Table 1:** *Result on CIFAR-10 dataset using C1, C2 network. We show the Hessian spectrum of different batch training models, and the corresponding performances on adversarial dataset generated by training/testing dataset (testing result is given in parenthese).*

|  | Batch | Acc. | $\lambda_1^\theta$ | $\lambda_1^{\mathbf{x}}$ | $\|\nabla_{\mathbf{x}}\mathcal{J}\|$ | Acc $\epsilon = 0.02$ | Acc $\epsilon = 0.01$ |
|---|---|---|---|---|---|---|---|
| C1 Cifar-10 | 16 | 100 (77.68) | 0.64 (32.78) | 2.69 (200.7) | 0.05 (20.41) | 48.07 (30.38) | 72.67 (42.70) |
|  | 32 | 100 (76.77) | 0.97 (45.28) | 3.43 (234.5) | 0.05 (23.55) | 49.04 (31.23) | 72.63 (43.30) |
|  | 64 | 100 (77.32) | 0.77 (48.06) | 3.14 (195.0) | 0.04 (21.47) | 50.40 (32.59) | 73.85 (44.76) |
|  | 128 | 100 (78.84) | 1.33 (137.5) | 1.41 (128.1) | 0.02 (13.98) | 33.15 (25.2) | 57.69 (39.09) |
|  | 256 | 100 (78.54) | 3.34 (338.3) | 1.51 (132.4) | 0.02 (14.08) | 25.33 (19.99) | 50.10 (34.94) |
|  | 512 | 100 (79.25) | 16.88 (885.6) | 1.97 (100.0) | 0.04 (10.42) | 14.17 (12.94) | 28.54 (25.08) |
|  | 1024 | 100 (78.50) | 51.67 (2372) | 3.11 (146.9) | 0.05 (13.33) | 8.80 (8.40) | 23.99 (21.57) |
|  | 2048 | 100 (77.31) | 80.18 (3769) | 5.18 (240.2) | 0.06 (18.08) | 4.14 (3.77) | 17.42 (16.31) |
| C2 Cifar-10 | 256 | 100 (79.20) | 0.62 (28) | 12.10 (704.0) | 0.10 (41.95) | 0.57 (0.38) | 0.73 (0.47) |
|  | 512 | 100 (80.44) | 0.75 (57) | 4.82 (425.2) | 0.03 (26.14) | 0.34 (0.25) | 0.54 (0.38) |
|  | 1024 | 100 (79.61) | 2.36 (142) | 0.523 (229.9) | 0.04 (17.16) | 0.27 (0.22) | 0.46 (0.35) |
|  | 2048 | 100 (78.99) | 4.30 (307) | 0.145 (260.0) | 0.50 (17.94) | 0.18 (0.16) | 0.33 (0.28) |

## 3 Large Batch, Adversarial Attack and Robust training

We first give a brief overview of adversarial attack and robust training and then present results connecting these with large batch size training.

### 3.1 Robust Optimization and Adversarial Attack

The methods for adversarial attack on a neural network can be broadly split into white-box attacks, where the model architecture and its parameters are known, and black-box attacks where such information is unavailable. Here we focus on the white-box methods, and in particular the optimization-based approach both for the attack and defense.

Suppose $\mathcal{M}(\theta)$ is a learning model (the neural network architecture), and $(\mathbf{x}, y)$ are the input data and the corresponding labels. The loss functional of the network with parameter $\theta$ on $(\mathbf{x}, y)$ is denoted by $\mathcal{J}(\theta, \mathbf{x}, y)$. For adversarial attack, we seek a perturbation $\Delta \mathbf{x}$ (with a bounded $L_\infty$ or $L_2$ norm) such that it maximizes $\mathcal{J}(\theta, \mathbf{x}, y)$:

$$\max_{\Delta \mathbf{x} \in \mathcal{U}} \mathcal{J}(\theta, \mathbf{x} + \Delta \mathbf{x}, y), \tag{1}$$

where $\mathcal{U}$ is an admissibility set for acceptable perturbation (typically restricting the magnitude of the perturbation). A typical choice for this set is $\mathcal{U} = \mathbf{B}(\mathbf{x}, \epsilon)$, a ball of radius $\epsilon$ centered at $\mathbf{x}$. A popular method for approximately computing $\Delta \mathbf{x}$, is Fast Gradient Sign Method [15], where the gradient of the loss functional is computed w.r.t. inputs, and the perturbation is set to:

$$\Delta \mathbf{x} = \epsilon \operatorname{sign}(\frac{\partial J(\mathbf{x}, \theta)}{\partial \mathbf{x}}) \tag{2}$$

This is not the only attack method possible. Other approaches include an iterative FGSM method (FGSM-10)[20] or using other norms such as $L_2$ norm instead of $L_\infty$ (We denote the $L_2$ method by $L_2 Grad$ in our results). Here we also use a second-order attack, where we use the Hessian w.r.t. input to precondition the gradient direction with second order information; Please see Table 5 in Appendix for details. One method to defend against such adversarial attacks, is to perform robust training [30, 22]:

$$\min_{\theta} \max_{\Delta \mathbf{x} \in \mathcal{U}} \mathcal{J}(\theta, \mathbf{x} + \Delta \mathbf{x}, y). \tag{3}$$

Solving this min-max optimization problem at each iteration requires first finding the worst adversarial perturbation that maximizes the loss, and then updating the model parameters $\theta$ for those cases. Since adversarial examples have to be generated at every iteration, it would not be feasible to find the exact perturbation that maximizes the objective function. Instead, a popular method is to perform a single or multiple gradient ascents to approximately compute $\Delta \mathbf{x}$. After computing $\Delta \mathbf{x}$ at each iteration, a typical optimization step (variant of SGD) is performed to update $\theta$.

Next we show that solving the maximization part is actually a saddle-free problem *almost everywhere*. This propert means the Hessian w.r.t input does not have negative eigenvalues which allows us to use CG for performing Newton solver for our second order adversarial perturbation tests in §3.4. [2]

### 3.2 Adversarial perturbation: A saddle-free problem

Recall that our loss functional is $\mathcal{J}(\theta; \mathbf{x}, y)$. We make following assumptions for the model to help show our theoretical result,

**Assumption 1.** *We assume the model's activation functions are strictly ReLu activation, and all layers are either convolution or fully connected. Here, Batch Normalization layers are accepted. Note that even though the ReLu activation has discontinuity at origin, i.e. $x = 0$, ReLu function is twice differentiable almost everywhere.*

The following theorem shows that the problem of finding an adversarial perturbation that maximized $\mathcal{J}$, is a saddle-free optimization problem, with a Positive-Semi-Definite (PSD) Hessian w.r.t. input almost everywhere. For details on the proof please see Appendix. A.1.

**Theorem 1.** *With Assumption. 1, for a DNN, its loss functional $\mathcal{J}(\boldsymbol{\theta}, \mathbf{x}, y)$ is a saddle-free function w.r.t. input $\mathbf{x}$ almost everywhere, i.e.*

$$\frac{\nabla^2 \mathcal{J}(\boldsymbol{\theta}, \mathbf{x}, y)}{\nabla \mathbf{x}^2} \succeq 0.$$

From the proof of Theorem 1, we could immediately get the following proposition of DNNs:

**Proposition 2.** *Based on Theorem 1 with Assumption 1, if the input $\mathbf{x} \in \mathbb{R}^d$ and the number of the output class is c, i.e. $y \in \{1, 2, 3 \ldots, c\}$, then the Hessian of DNNs w.r.t. to $\mathbf{x}$ is almost a rank c matrix almost everywhere; see Appendix A.1 for details.*

### 3.3 Large Batch Training and Robustness

Here, we test the robustness of the models trained with different batches to an adversarial attack. We use Fast Gradient Sign Method for all the experiments (we did not see any difference with FGSM-10 attack). The adversarial performance is measured by the fraction of correctly classified

**Table 2:** *Result on CIFAR-100 dataset using CR network. We show the Hessian spectrum of different batch training models, and the corresponding performances on adversarial dataset generated by training/testing dataset (testing result is given in parenthese).*

| Batch | Acc. | $\lambda_1^\theta$ | Acc $\epsilon = 0.02$ | Acc $\epsilon = 0.01$ |
|---|---|---|---|---|
| 64 | 99.98 (70.81) | 0.022 (10.43) | 61.54 (34.48) | 78.57 (39.94) |
| 128 | 99.97 (70.9 ) | 0.055 (26.50 ) | 58.15 (33.73) | 77.41 (38.77) |
| 256 | 99.98 (68.6 ) | 1.090 (148.29) | 39.96 (28.37) | 66.12 (35.02) |
| 512 | 99.98 (68.6 ) | 1.090 (148.29) | 40.48 (28.37) | 66.09 (35.02) |

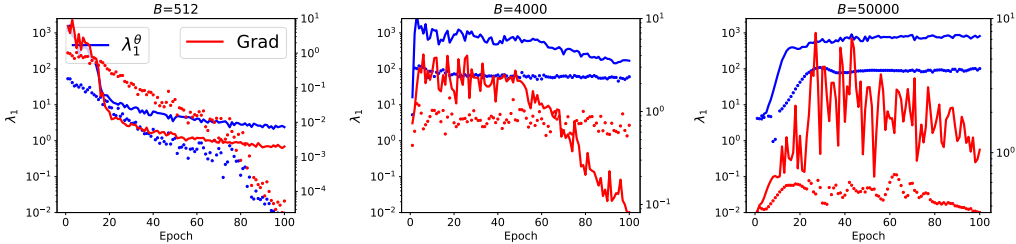

**Figure 4:** *Changes in the dominant eigenvalue of the Hessian w.r.t weights and the total gradient is shown for different epochs during training. Note the increase in $\lambda_0$ (blue curve) for large batch v.s. small batch. In particular, note that the values for total gradient along with the Hessian spectrum show that large batch does not get "stuck" in saddle points, but areas in the optimization landscape that have high curvature. More results are shown in Fig. 16. The dotted points show the corresponding results when using robust optimization, which makes the solver stay in areas with smaller spectrum.*

adversarial inputs. We report the performance for both the training and test datasets for different values of $\epsilon = 0.02, 0.01$ ($\epsilon$ is the metric for the adversarial perturbation magnitude in $L_\infty$ norm). The performance results for C1, and C2 models on CIFAR-10, CR model on CIFAR-100, are reported in the last two columns of Tables 1,and 2 (MNIST results are given in appendix, Table 7). The interesting observation is that for all the cases, large batches are considerably more prone to adversarial attacks as compared to small batches. This means that not only the model design affects the robustness of the model, but also the hyper-parameters used during optimization, and in particular the properties of the point that the model has converged to.

**Table 3:** *Accuracy of different models across different adversarial samples of MNIST, which are obtained by perturbing the original model $\mathcal{M}_{ORI}$*

| | $\mathcal{D}_{clean}$ | $\mathcal{D}_{FGSM}$ | $\mathcal{D}_{FGSM10}$ | $\mathcal{D}_{L_2GRAD}$ | $\mathcal{D}_{FHSM}$ | $\mathcal{D}_{L_2HESS}$ | MEAN of Adv |
|---|---|---|---|---|---|---|---|
| $\mathcal{M}_{ORI}$ | 99.32 | 60.37 | 77.27 | 14.32 | 82.04 | 33.21 | 53.44 |
| $\mathcal{M}_{FGSM}$ | 99.49 | 96.18 | 97.44 | 63.46 | 97.56 | 83.33 | 87.59 |
| $\mathcal{M}_{FGSM10}$ | **99.5** | 96.52 | **97.63** | 66.15 | 97.66 | 84.64 | 88.52 |
| $\mathcal{M}_{L_2GRAD}$ | 98.91 | **96.88** | 97.39 | **86.23** | **97.66** | **92.56** | **94.14** |
| $\mathcal{M}_{FHSM}$ | 99.45 | 94.41 | 96.48 | 52.67 | 96.89 | 77.58 | 83.60 |
| $\mathcal{M}_{L_2HESS}$ | 98.72 | 95.02 | 96.49 | 77.18 | 97.43 | 90.33 | 91.29 |

From this result, there seems to be a strong correlation between the spectrum of the Hessian w.r.t. $\theta$ and how robust the model is. However, we want to emphasize that in general there is no correlation between the Hessian w.r.t. weights and the robustness of the model w.r.t. the input. For instance, consider a two variable function $\mathcal{J}(\boldsymbol{\theta}, \mathbf{x})$ (we treat $\theta$ and $\mathbf{x}$ as two single variables), for which the Hessian spectrum of $\theta$ has no correlation to robustness of $\mathcal{J}$ w.r.t. $\mathbf{x}$. This can be easily demonstrated for a least squares problem, $L = \|\theta \mathbf{x} - \mathbf{y}\|_2^2$. It is not hard to see the Hessian of $\theta$ and $\mathbf{x}$ are, $\mathbf{x}\mathbf{x}^T$ and $\theta\theta^T$, respectively. Therefore, in general we cannot link the Hessian spectrum w.r.t. weights to robustness of the network. However, the numerical results for all the neural networks show that models that have higher Hessian spectrum w.r.t. $\theta$ are also more prone to adversarial attacks. A potential explanation for this would be to look at how the gradient and Hessian w.r.t. input (i.e. $\mathbf{x}$) would change for different batch sizes. We have computed the dominant eigenvalue of this Hessian using power iteration for each individual input sample for both training and testing datasets. Furthermore, we have computed the norm of the gradient w.r.t. $\mathbf{x}$ for these datasets as well. These two

**Table 4:** *Accuracy of different models across different samples of CIFAR-10, which are obtained by perturbing the original model $\mathcal{M}_{ORI}$*

| | $\mathcal{D}_{clean}$ | $\mathcal{D}_{FGSM}$ | $\mathcal{D}_{FGSM10}$ | $\mathcal{D}_{L_2GRAD}$ | $\mathcal{D}_{FHSM}$ | $\mathcal{D}_{L_2HESS}$ | MEAN of Adv |
|---|---|---|---|---|---|---|---|
| $\mathcal{M}_{ORI}$ | **79.46** | 15.25 | 4.46 | 12.37 | 29.64 | 22.93 | 16.93 |
| $\mathcal{M}_{FGSM}$ | 71.82 | 63.05 | 63.44 | 57.68 | **66.04** | 62.36 | 62.51 |
| $\mathcal{M}_{FGSM10}$ | 71.14 | **63.32** | **63.88** | **58.25** | 65.95 | **62.70** | **62.82** |
| $\mathcal{M}_{L_2GRAD}$ | 63.52 | 59.33 | 59.73 | 57.35 | 60.44 | 58.98 | 59.16 |
| $\mathcal{M}_{FHSM}$ | 74.34 | 47.65 | 43.95 | 38.45 | 62.75 | 55.77 | 49.71 |
| $\mathcal{M}_{L_2HESS}$ | 71.59 | 50.05 | 46.66 | 42.95 | 62.87 | 58.42 | 52.19 |

metrics are reported in $\lambda_1^{\mathbf{x}}$, and $\|\nabla_x \mathcal{J}\|$; See Table 1 for details. The results on all of our experiments show that these two metrics actually do not correlate with the adversarial accuracy. For instance, consider C1 model with $B = 512$. It has both smaller gradient and smaller Hessian eigenvalue w.r.t. $\mathbf{x}$ as compared to $B = 32$, but it performs note acidly worse under adversarial attack. One possible reason for this could be that the decision boundaries for large batches are less stable, such that with small adversarial perturbation the model gets fooled.

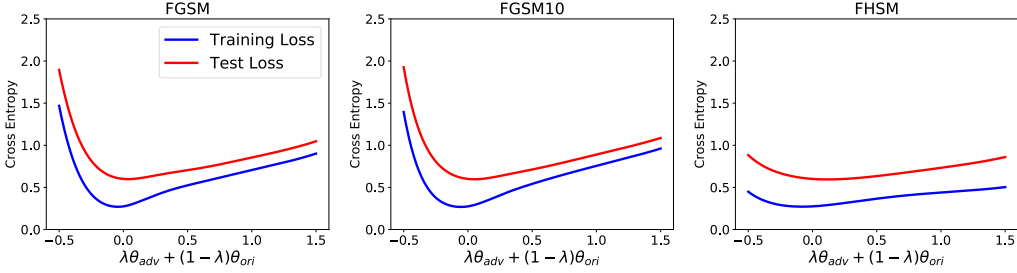

**Figure 5:** *1-D Parametric plot for C3 model on CIFAR-10. We interpolate between parameters of $\mathcal{M}_{ORI}$ and $\mathcal{M}_{ADV}$, and compute the cross entropy loss on the y-axis.*

### 3.4 Adversarial Training and Hessian Spectrum

In this part, we study how the Hessian spectrum and the landscape of the loss functional change after adversarial training is performed. Here, we fix the batch size (and all other optimization hyper-parameters) and use five different adversarial training methods as described in §3.1.

For the sake of clarity let us denote $\mathcal{D}$ to be the test dataset which can be the original clean test dataset or one created by using an adversarial method. For instance, we denote $\mathcal{D}_{FGSM}$ to be the adversarial dataset generated by FGSM, and $\mathcal{D}_{clean}$ to be the original clean test dataset.

**setup:** For the MNIST experiments, we train a standard LeNet on MNIST dataset [3] (using M1 network). For the original training, we set the learning rate to 0.01 and momentum to 0.9, and decay the learning rate by half after every 5 epochs, for a total of 100 epochs. Then we perform an additional five epochs of adversarial training with a learning rate of $0.01$. The perturbation magnitude, $\epsilon$, is set to $0.1$ for $L_\infty$ attack and $2.8$ for $L_2$ attack. We also present results for C3 model [4] on

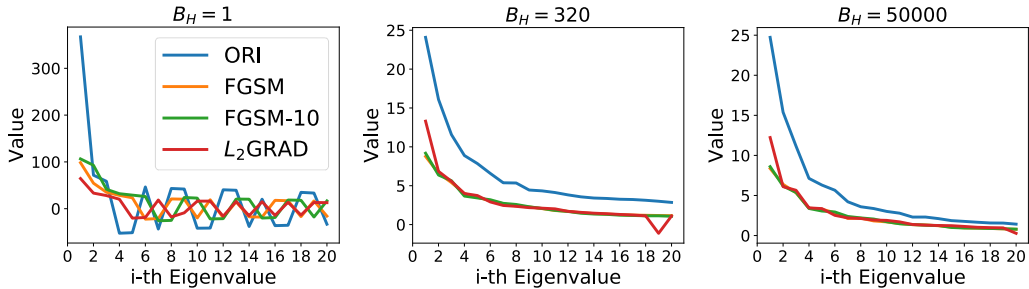

**Figure 6:** *Spectrum of the sub-sampled Hessian of the loss functional w.r.t. weights. The results are computed for different batch sizes, which are randomly chosen, of $B = 1$, 320, 50000 of C1.*

CIFAR-10, using the same hyper-parameters, except that the training is performed for 100 epochs. Afterwards, adversarial training is performed for a subsequent 10 epochs with a learning rate of 0.01 and momentum of 0.9 (the learning rate is decayed by half after five epochs). Furthermore, the adversarial perturbation magnitude is set to $\epsilon = 0.02$ for $L_\infty$ attack and 1.2 for $L_2$ attack[27].

The results are shown in Table 3, 4. We can see that after adversarial training the model becomes more robust to these attacks. Note that the accuracy of different adversarial attacks varies, which is expected since the various strengths of different attack method. In addition, all adversarial training methods improve the robustness on adversarial dataset, though they lose some accuracy on $\mathcal{D}_{clean}$, which is consistent with the observations in [15]. As an example, consider the second row of Table 3 which shows the results when FGSM is used for robust training. The performance of this model when tested against the $L_2GRAD$ attack method is 63.46% as opposed to 14.32% of the original model ($\mathcal{M}_{ORI}$). The rest of the rows show the results for different algorithms.

The main question here is how the landscape of the loss functional is changed after these robust optimizations are performed? We first show a 1-D parametric interpolation between the original model parameters $\theta$ and that of the robustified models, as shown in Fig. 5 (see Fig. 11 for all cases) and 10. Notice the robust models are at a point that has smaller curvature as compared to the original model. To exactly quantify this, we compute the spectrum of the Hessian as shown in Fig. 6, and 12. Besides the full Hessian spectrum, we also report the spectrum of sub-sampled Hessian. The latter is computed by randomly selecting a subset of the training dataset. We denote the size of this subset as $B_H$ to avoid confusion with the training batch size. In particular, we report results for $B_H = 1$ and $B_H = 320$. There are several important observations here. First, notice that the spectrum of the robust models is noticeably smaller than the original model. This means that the min-max problem of Eq. 3 favors areas with lower curvature. Second, note that even though the total Hessian shows that we have converged to a point with positive curvature (at least based on the top 20 eigenvalues), but that is not necessarily the case when we look at individual samples (i.e. $B_H = 1$). For a randomly selected batch of $B_H = 1$, we see that we have actually converged to a point that has both positive and negative curvatures, with a non-zero gradient (meaning it is not a saddle point). To the best of our knowledge this is a new finding, but one that is expected as SGD optimizes the expected loss instead of individual ones.

Now going back to Fig. 4, we show how the spectrum changes during training when we use robust optimization. We can clearly see that with robust optimization the solver is pushed to areas with smaller spectrum as opposed to when we do not use robust training. This is a very interesting finding and shows the possibility of using robust training as a systematic means to bias the solver to avoid *sharp* minimas.

## 4  Conclusion

We studied Neural Networks through the lens of the Hessian operator. In particular, we studied large batch size training and its connection with stability of the model in the presence of white-box adversarial attacks. By computing Hessian spectrum, we provided several evidences that show that large batch size training tends to get attracted to areas with higher Hessian spectrum. We reported the eigenvalues of the Hessian w.r.t. whole dataset, and plotted the landscape of the loss when perturbed along the dominant eigenvector. Visual results were in line with the numerical values for the spectrum. Our empirical results show that adversarial attacks/training and large batches are closely related. We provided several empirical results on multiple datasets that show large batch size training is more prone to adversarial attacks (more results are provided in the supplementary material). This means that not only the model design is important, but also the optimization hyper-parameters can drastically affect a network's robustness. Furthermore, we observed that robust training is antithetical to large batch size training, in the sense that it favors areas with noticeably smaller Hessian spectrum w.r.t. $\theta$.

The results show that the robustness of the model does not (at least directly) correlate with the Hessian w.r.t. $\mathbf{x}$. We also found that this Hessian is actually a PSD matrix, meaning that the problem of finding the adversarial perturbation is actually a saddle-free problem for cases that satisfy assumption 1. Furthermore, we showed that even though the model may converge to an area with positive curvature when considering all of the training dataset (i.e. total loss), but if we look at individual samples then the Hessian can actually have significant negative eigenvalues. From an optimization viewpoint, this is due to the fact that SGD optimizes the expected loss and not the individual per sample loss.

# 5 Rebuttal

We would like to thank all the reviewers and area chair for taking the time to review our work and providing us with their valuable feedback. Below we discuss the main comments:

**Table 5:** *Result on SVHN dataset using C1. We use the full training dataset with 530K images. The large batch size behavior is consistent with other datasets (i.e. CIFAR-10, CIFAR-100, and MNIST).*

| Batch | Acc | $\lambda_1^\theta$ | Acc $\epsilon = 0.05$ | Acc $\epsilon = 0.02$ |
|---|---|---|---|---|
| 256 | 100 (95.70 ) | 1.85 (87.71) | 23.59 (16.46) | 50.96 (35.89) |
| 1024 | 100 (95.41 ) | 18.23 (184.7) | 16.65 (11.88) | 42.67 (29.27) |
| 4096 | 100 (95.22 ) | 58.46 (606.1) | 8.36 (6.33) | 27.04 (17.78) |
| 16384 | 100 (94.86 ) | 74.28 (1040) | 6.25 (4.95) | 22.31 (15.43) |

∗**Preliminary results for adversarial regularization**

(A) Following this paper, we have designed a new adaptive algorithm that uses adversarial training (robust optimization) in combination with second order information that achieves state-of-the-art performance for large batch training (please see [33]). The main goal of this work has been to perform detailed analysis to better understand the problems with large batch training.

∗ **ReLU has 0 Hessian a.e. and I suggest adding analysis with twice differentiable activation.**

(A) This is an excellent observation regarding ReLU networks. We have performed new experiments with the suggested activation functions (Softplus and ELU) and show results in Table 6. The reason we chose ReLU activation was that many/most of the new neural networks are incorporating it. However, our results still hold for twice differentiable activations as well. That is, larger batches are less robust (please see last two columns of Table 6) and get attracted to areas with higher Hessian spectrum, also known as sharper points, in the optimization space. We have also visually plotted the dominant eigenvalue of the Hessian versus batch size for different activation functions in Figure 7. This clearly shows the same trend. We will add these results to the final version of the paper.

∗ **Experiments on two small datasets MNIST and CIFAR-10.**

(A) Our results are not limited to these two small datasets. We have addressed this fair concern of the reviewer by running an experiment on full SVHN dataset with 530K images as shown in Table 5. We can see the results are consistent with the other datasets in that larger batches are less robust to adversarial perturbation (last two columns), and the Hessian spectrum also increases for larger batch (please see $\lambda_1^\theta$ column).

| | Batch | Acc. | $\lambda_1^\theta$ | Acc $\epsilon = 0.02$ |
|---|---|---|---|---|
| **C1S** | 128 | 100.00 (78.79 ) | 4.45 (318.9) | 20.40 (17.79) |
| | 256 | 100.00 (78.79 ) | 5.00 (507.4) | 17.79 (16.19) |
| | 512 | 100.00 (78.68 ) | 16.18 (819.2) | 12.99 (11.55) |
| | 1024 | 100.00 (77.78 ) | 46.99 (2030) | 5.55 (5.42) |
| | 2048 | 100.00 (76.27 ) | 97.71 (4329) | 2.38 (2.29) |
| **C1E** | 128 | 100.00 (78.94 ) | 4.32 (271.4) | 17.37 (15.26) |
| | 256 | 100.00 (78.88 ) | 17.39 (469.2) | 13.44 (12.01) |
| | 512 | 100.00 (78.38 ) | 27.23 (1048) | 9.20 (8.74) |
| | 1024 | 100.00 (77.82 ) | 62.64 (2392) | 4.10 (3.99) |
| | 2048 | 100.00 (76.64 ) | 114.4 (4347) | 1.55 (1.6) |

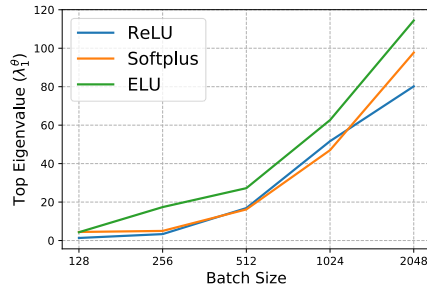

**Table 6:** *Results on CIFAR-10 dataset by C1S (replace all ReLU by Softplus, $\beta = 20$) and C1E (replace all ReLU by ELU, $\alpha = 1$). We see the same trend with these twice differentiable activations as with ReLU.*

**Figure 7:** *Top eigenvalue on training dataset for C1 with different activation functions on various batch size.*

## Footnotes

[2]This results might also be helpful for finding better optimization strategies for GANS.

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
