[Supplementary Material · nips18-appendix.pdf]

# 6 Appendix

## 6.1 Proof of Theorem 1

In this section, we give the proof of Theorem 1. The first thing we want to point out is that, although we prove the Hessians of these NNs are positive semi-definite almost everywhere, these NNs are not convex w.r.t. inputs, i.e., $\mathbf{x}$. The discontinuity of ReLU is the cause. (For instance, consider a combination of two step functions in 1-D, e.g. $f(x) = 1_{x \geq 1} + 1_{x \geq 2}$ is not a convex function but has $0$ second derivative almost everywhere.) However, this has an important implication, that the problem is saddle-free.

Before we go to the proof of Theorem 1, let us prove the following lemma for cross-entropy loss with soft-max layer.

**Lemma 3.** *Let us denote by $\mathbf{s} \in \mathbb{R}^d$ the input of the soft-max function, by $y \in \{1, 2, \ldots, d\}$ the correct label of the inputs $\mathbf{x}$, by $g(\mathbf{s})$ the soft-max function, and by $L(\mathbf{s}, y)$ the cross-entropy loss. Then we have*

$$\frac{\partial^2 L(\mathbf{s}, y)}{\partial \mathbf{s}^2} \succeq \mathbf{0}.$$

*Proof.* Let $s_d = \sum_{j=1}^{d} e^{\mathbf{s}_j}$, $\mathbf{p}_i = \frac{e^{\mathbf{s}_i}}{s_d}$, and then it follows that

$$L(\mathbf{s}, y) = -\sum_{i=1}^{d} y_i \log \mathbf{p}_i.$$

Further, it is not hard to see that

$$\frac{\partial L(\mathbf{s}, y)}{\partial \mathbf{s}_j} = -\sum_{i=1}^{d} y_i \frac{\partial \log \mathbf{p}_i}{\partial \mathbf{s}_j}$$

$$= -y_j(1 - \mathbf{p}_j) - \sum_{i \neq j} y_i \frac{\mathbf{p}_k \mathbf{p}_j}{\mathbf{p}_k}$$

$$= \mathbf{p}_j - y_j.$$

Then, the second order derivative of $L$ w.r.t. $\mathbf{s}_i \mathbf{s}_j$ is

$$\frac{\partial^2 L(\mathbf{s}, y)}{\partial \mathbf{s}_j^2} = \mathbf{p}_j(1 - \mathbf{p}_j), \quad \text{and} \quad \frac{\partial^2 L(\mathbf{s}, y)}{\partial \mathbf{s}_j \partial \mathbf{s}_i} = -\mathbf{p}_j \mathbf{p}_i.$$

Since

$$\frac{\partial^2 L(\mathbf{s}, y)}{\partial \mathbf{s}_j^2} + \sum_{i \neq j} \frac{\partial^2 L(\mathbf{s}, y)}{\partial \mathbf{s}_j \partial \mathbf{s}_i} = 0, \quad \text{and} \quad \frac{\partial^2 L(\mathbf{s}, y)}{\partial \mathbf{s}_j^2} \geq 0,$$

we have

$$\frac{\partial^2 L(\mathbf{s}, y)}{\partial \mathbf{s}^2} \succeq \mathbf{0}.$$

$\square$

Now, let us give the proof of Theorem 1:

Assume the input of the soft-max layer is $\mathbf{s}$ and the cross-entropy is $L(\mathbf{s}, y)$. Based on Chain Rule, it follows that

$$\frac{\partial \mathcal{J}(\boldsymbol{\theta}, \mathbf{x}, y)}{\partial \mathbf{x}} = \frac{\partial L}{\partial \mathbf{s}} \frac{\partial \mathbf{s}}{\partial \mathbf{x}}.$$

From Assumption. 1 we know that all the layers before the soft-max are either linear or ReLU, which indicates $\frac{\partial^2 \mathbf{s}}{\partial \mathbf{x}^2} = \mathbf{0}$ (a tensor) almost everywhere. Therefore, applying chain rule again for the above equation,

$$\frac{\partial^2 \mathcal{J}(\boldsymbol{\theta}, \mathbf{x}, y)}{\partial \mathbf{x}^2} = (\frac{\partial \mathbf{s}}{\partial \mathbf{x}})^T \frac{\partial^2 L}{\partial \mathbf{s}^2} \frac{\partial \mathbf{s}}{\partial \mathbf{x}} + \frac{\partial L}{\partial \mathbf{s}} \frac{\partial^2 \mathbf{s}}{\partial \mathbf{x}^2}$$

$$= (\frac{\partial \mathbf{s}}{\partial \mathbf{x}})^T \frac{\partial^2 L}{\partial \mathbf{s}^2} \frac{\partial \mathbf{s}}{\partial \mathbf{x}}.$$

It is easy to see $\dfrac{\partial^2 \mathcal{J}(\boldsymbol{\theta}, \mathbf{x}, y)}{\partial \mathbf{x}^2} \succeq 0$ almost everywhere since $\dfrac{\partial^2 L}{\partial \mathbf{s}^2} \succeq 0$ from Lemma 3.

From above we could see that the Hessian of NNs w.r.t. $\mathbf{x}$ is at most a rank $c$ (the number of class) matrix, since the rank of the Hessian matrix

$$\frac{\partial^2 \mathcal{J}(\boldsymbol{\theta}, \mathbf{x}, y)}{\partial \mathbf{x}^2} = (\frac{\partial \mathbf{s}}{\partial \mathbf{x}})^T \frac{\partial^2 L}{\partial \mathbf{s}^2} \frac{\partial \mathbf{s}}{\partial \mathbf{x}}$$

is dominated by the term $\frac{\partial^2 L}{\partial \mathbf{s}^2}$, which is at most rank $c$.

## 6.2 Attacks Mentioned in Paper

In this section, we show the details about the attacks used in our paper. Please see Table 7 for details.

**Table 7:** *The definition of all attacks used in the paper. Here* $\mathbf{g}_x \triangleq \dfrac{\partial \mathcal{J}(\mathbf{x}, \theta)}{\partial \mathbf{x}}$ *and* $\mathbf{H}_x \triangleq \dfrac{\partial^2 \mathcal{J}(\mathbf{x}, \theta)}{\partial \mathbf{x}^2}$.

|  | $\Delta \mathbf{x}$ |
|---|---|
| FGSM | $\epsilon \cdot \mathrm{sign}(\mathbf{g}_x)$ |
| FGSM-10 | $\epsilon \cdot \mathrm{sign}(\mathbf{g}_x)$ (iterate 10 times) |
| $L_2$GRAD | $\epsilon \cdot \mathbf{g}_x / \|\mathbf{g}_x\|$ |
| FHSM | $\epsilon \cdot \mathrm{sign}(\mathbf{H}_x^{-1}\mathbf{g}_x)$ |
| $L_2$HESS | $\epsilon \cdot \mathbf{H}_x^{-1}\mathbf{g}_x / \|\mathbf{H}_x^{-1}\mathbf{g}_x\|$ |

## 6.3 Models Mentioned in Paper

In this section, we give the details about the NNs used in our paper. For clarification, We omit the ReLu activation here. However, in practice, we implement ReLu regularity. Also, for all convolution layers, we add padding to make sure there is no dimension reduction. We denote Conv(a,a,b) as a convolution layer having b channels with a by a filters, MP(a,a) as a a by a max-pooling layer, FN(a) as a fully-connect layer with a output and SM(a) is the soft-max layer with a output. For our Conv(5,5,b) (Conv(3,3,b))layers, the stride is 2 (1). See Table 8 for details of all models used in this paper.

**Table 8:** *The definition of all models used in the paper.*

| Name | Structure |
|---|---|
| C1 (for CIFAR-10) | Conv(5,5,64) – MP(3,3) – BN–Conv(5,5,64) –MP(3,3)–BN–FN(384)–FN(192)–SM(10) |
| C2 (for CIFAR-10) | Conv(3,3,63)–BN–Conv(3,3,64)–BN–Conv(3,3,128) –BN–Conv(3,3,128)–BN–FC(256)–FC(256)–SM(10) |
| C3 (for CIFAR-10) | Conv(3,3,64)–Conv(3,3,64)–Conv(3,3,128) –Conv(3,3,128)–FC(256)–FC(256)–SM(10) |
| M1 (for MNIST) | Conv(5,5,20)–Conv(5,5,50)–FC(500)–SM(10) |
| CR (for CIFAR-100) | ResNet18 For CIFAR-100 |

## 6.4 Discussion on Second Order Method

Although second order adversarial attack looks well for MNIST (see Table 3), but for most our experiments on CIFAR-10 (see Table 4), the second order methods are weaker than variations of the gradient based methods. Also, notice that the robust models trained by second order method are also more prone to attack on CIFAR-10, particularly $\mathcal{M}_{FHSM}$ and $\mathcal{M}_{L_2HESS}$. We give two potential explanation here.

First note that the Hessian w.r.t. input is a low rank matrix. In fact, as mentioned above, the rank of the input Hessian for CIFAR-10 is at most ten; see Proposition 2, the matrix itself is $3K \times 3K$. Even though we use inexact Newton method [10] along with Conjugate Gradient solver, but this low rank nature creates numerical problems. Designing preconditioners for second-order attack is part of our

future work. The second point is that, as we saw in the previous section the input Hessian does not directly correlate with how robust the network is. In fact, the most effective attack method would be to perturb the input towards the decision boundary, instead of just maximizing the loss.

## 6.5 More Numerical Result for §2 and 3

In this section, we provide more numerical results for §2 and 3. All conclusions from the numerical results are consistency with those in §2 and 3.

**Table 9:** *Result on MNIST dataset for M1 model (LeNet-5). We shows the Hessian spectrum of different batch training models, and the corresponding performances on adversarial dataset generated by training/testing dataset. The testing results are shown in parenthesis. We report the adversarial accuracy of three different magnitudes of attack. The interesting observation is that the $\lambda_1^\theta$ is increasing while the adversarial accuracy is decreasing for fixed $\epsilon$. Meanwhile, we do not know if there is a relationship between $\lambda_1^\theta$ and Clean accuracy or not. Also, we cannot see the relation between $\lambda_1^{\mathbf{x}}$, $\|\nabla_{\mathbf{x}}\mathcal{J}(\theta, \mathbf{x}, y)\|$ and the adversarial accuracy.*

| Batch | Acc | $\lambda_1^\theta$ | $\lambda_1^{\mathbf{x}}$ | $\|\partial_{\mathbf{x}}\mathcal{J}(\theta, \mathbf{x}, y)\|$ | Acc $\epsilon = 0.2$ | Acc $\epsilon = 0.1$ |
|---|---|---|---|---|---|---|
| 64 | 100 (99.21) | 0.49 (2.96 ) | 0.07 (0.41) | 0.007 (0.10) | 0.53 (0.53) | 0.85 (0.85) |
| 128 | 100 (99.18) | 1.44 (8.10 ) | 0.10 (0.51) | 0.009 (0.12) | 0.50 (0.51) | 0.83 (0.83) |
| 256 | 100 (99.04) | 2.71 (13.54) | 0.09 (0.50) | 0.008 (0.12) | 0.45 (0.46) | 0.81 (0.82) |
| 512 | 100 (99.04) | 5.84 (26.35) | 0.12 (0.52) | 0.010 (0.13) | 0.42 (0.42) | 0.79 (0.80) |
| 1024 | 100 (99.05) | 21.24 (36.96) | 0.25 (0.42) | 0.032 (0.11) | 0.32 (0.33) | 0.73 (0.74) |
| 2048 | 100 (98.99) | 44.30 (49.36) | 0.36 (0.39) | 0.075 (0.11) | 0.19 (0.19) | 0.72 (0.73) |

**Figure 8:** *The landscape of the loss functional is shown along the dominant eigenvector of the Hessian on MNIST for M1. Note that the $y-axis$ is in logarithm scale. Here $\epsilon$ is a scalar that perturbs the model parameters along the dominant eigenvector denoted by $v_1$. The green line is the loss for a randomly batch with batch-size 320 on MNIST. The blue and red line are the training and test loss, respectively. From the figure we could see that the curvature of test loss is much larger than training.*

**Figure 9:** *We show the landscape of the test and training objective functional along the first eigenvector of the sub-sampled Hessian with $B = 320$, i.e. 320 samples from training dataset, on MNIST for M1. We plot both the batch loss as well as the total training and test loss. One can see that visually the results show that the robust models converge to a region with smaller curvature.*

**Figure 10:** *We show the landscape of the test and training objective functional along the first eigenvector of the sub-sampled Hessian with $B = 320$, i.e. 320 samples from training, on CIFAR-10 for C3. We plot both the batch loss as well as the total training and test loss. One can see that visually the results show that the curvature of robust models is smaller.*

**Figure 11:** *1-D Parametric Plot on MNIST for M1 of $\mathcal{M}_{ORI}$ and adversarial models. Here we are showing how the landscape of the total loss functional changes when we interpolate from the original model ($\lambda = 0$) to the robust model ($\lambda = 1$). For all cases the robust model ends up at a point that has relatively smaller curvature compared to the original network.*

**Figure 12:** *1-D Parametric Plot on CIFAR-10 of $\mathcal{M}_{ORI}$ and adversarial models, i.e. total loss functional changes interpolating from the original model ($\lambda = 0$) to the robust model ($\lambda = 1$). For all cases the robust model ends up at a point that has relatively smaller curvature compared to the original network.*

**Figure 13:** *Spectrum of the sub-sampled Hessian of the loss functional w.r.t. the model parameters computed by power iteration on MNIST of M1. The results are computed for different batch sizes of $B = 1$, $B = 320$, and $B = 60000$. We report two cases for the single batch experiment, which is drawn randomly from the clean training data. The results show that the sub-sampled Hessian spectrum decreases for robust models. An interesting observation is that for the MNIST dataset, the original model has actually converged to a saddle point, even though it has a good generalization error. Also notice that the results for $B = 320$ and $B = 60,000$ are relatively close, which hints that the curvature for the full Hessian should also be smaller for the robust methods. This is demonstrated in Fig. 9.*

**Table 10:** *Baseline accuracy is shown for large batch size for C1 model along with results aciheved with scaling learning rate method proposed by [16] (denoted by "FB Acc"). The last column shows results when training is performed with robust optimization. As we can see, the performance of the latter is actually better for large batch size. We emphasize that the goal is to perform analysis to better understand the problems with large batch size training. More extensive tests are needed before one could claim that robust optimization performs better than other methods.*

| Batch | Baseline Acc | FB Acc | Robust Acc |
|-------|-------------|--------|-----------|
| 8000  | 0.7559      | 0.752  | 0.7612    |
| 10000 | 0.7561      | 0.1    | 0.7597    |
| 25000 | 0.7023      | 0.1    | 0.7409    |
| 50000 | 0.5523      | 0.1    | 0.7116    |

**Figure 14:** *The landscape of the loss functional is shown when the C2 model parameters are changed along the first two dominant eigenvectors of the Hessian. Here $\epsilon_1$, $\epsilon_2$ are scalars that perturbs the model parameters along the first and second dominant eigenvectors.*

**Figure 15:** *The landscape of the loss functional is shown when the M1 model parameters are changed along the first two dominant eigenvectors of the Hessian. Here $\epsilon_1$, $\epsilon_2$ are scalars that perturbs the model parameters along the first and second dominant eigenvectors.*

**Figure 16:** *The landscape of the loss functional is shown along the dominant eigenvector of the Hessian for C2 architecture on CIFAR-10 dataset. Here $\epsilon$ is a scalar that perturbs the model parameters along the dominant eigenvector denoted by $v_1$.*

**Figure 17:** *Changes in the dominant eigenvalue of the Hessian w.r.t weights and the total gradient is shown for different epochs during training. Note the increase in $\lambda_1^\theta$ (blue curve) for large batch vs small batch. In particular, note that the values for total gradient along with the Hessian spectrum show that large batch does not get "stuck" in saddle points, but areas in the optimization landscape that have high curvature. The dotted points show the corresponding results when we use robust optimization. We can see that this pushes the training to flatter areas. This clearly demonstrates the potential to use robust optimization as a means to avoid sharp minimas.*