[Reviews · NeurIPS 2018]

Reviewer 1



Summary: This paper presents many empirical results on the effects of large batch training based on second order analysis. The results focus on the generalization gap and robustness to adversaries under the setting of large batch training. The first major conclusion is that large batch size training tends to get attracted to areas with higher Hessian spectrum, rather than saddle points. Secondly, large batch training drastically reduces model robustness, and robust training favors parameters with smaller Hessian spectrum. Strength: This paper presents many intriguing results under the setting of large batch training. Experiments are comprehensive. Despite lacking theoretical support, the results in this paper help us understand large batch training better and develop new theoretical frameworks. This is also a pioneering result on empirically investigating Hessian-based analysis to understand large batch training, especially on its connections to model robustness. The presentation of the paper is nice, with many nicely drawn figures and tables. The paper is overall well written. Good job! Empirical studies in this paper are also done very carefully with sufficient details presented. So I am confident that the experimental results can be reproduced by others. Weakness: Despite the Hessian based analysis shows some intriguing properties of the networks under investigation, all experiments in this paper use ReLU networks. In fact, the only contributing term in Hessian of the network comes from the softmax layer (Appendix line 388), which does not contain any trainable parameter. Removing the softmax layer, the network still works as a valid classifier (and we pick the class with largest logit value as the prediction result), but the analysis in this paper would fail as the Hessian is 0 almost everywhere. Given this fact, I am in doubt if the results discovered in this paper actually reflects the property of ReLU networks. In fact, the power of the ReLU network comes from the singular points where Hessian does not exists, which provides non-linearity; without these points the network would become a pure linear network. I would suggest that conducting empirical analysis on networks with twice-differentiable activations, for example, ELU or softplus, to give more convincing conclusions. Especially, softplus function can approximate ReLU in its limit. The authors can change the scaling parameter in softplus to simulate the case where the Hessian is close to the one of ReLU networks, and compare this result to the existing results presented in this draft. The theoretical contribution of this paper is also weak; the PSD property of Hessian matrices comes from a simple fact the Hessian of softmax function is PSD. Overall Evaluation: I appreciate the comprehensive experiments, detailed experiment setup, and intriguing findings in this paper. Empirical results are interesting, but are not completely sound theoretically (see the weakness about). Overall I am leaning slightly towards an rejection, but accepting it would not be that bad. Comments after author feedback: I have carefully read the author feedback. I am glad that the authors conduct additional experiments on twice differentiable activation functions and provide some supporting evidence. I believe the results of this paper are basically correct (despite using only ReLU), but I think the main texts should include more experiments using twice differentiable networks to make the conclusions more convincing (and this requires a relatively large change to this paper). I am fine if the AC wants to accept this paper, and I hope the authors can revise the paper accordingly if accepted.

Reviewer 2



Large batch sizes have only received attention recently (with most papers published in 2018), motivated by GPU parallelization and the observation that increasing the batch size may work as well or better than decreasing the step size. Observations are still contradictory, but this paper represents the best attempt so far to bring some clarity to a very messy situation. This paper is superbly written with extensive experiments. The main limit to clarity is the authors desire to be as complete as possible (constantly sending the reader to figures which are in the appendix, and whose captions are unsufficiently explanatory) and overcautious in drawing conclusions from their experiments. Its density may make it hard to access for readers who are not yet familiarized with the problem. This paper observes, through extensive experiments on 2 datasets, that models trained with large batch sizes, even when their test performance is comparable (or slightly inferior) to models trained with smaller batches, and considerably more sensitive to adversarial attacks at test time. However, if adversarial noise is added at train time, performance with large batch size can be kept high, and maybe even become better than smaller batches. The authors only have some preliminary result for this (in the appendix!), and are not providing a ‘large batch’ recipe yet. The main technical contribution is in how adversarial perturbation techniques (5 of them are covered in this paper) enable to control the sharpness of the Hessian (measured with the eigenvalues). Does the noise need to be adversarial in practice? I am not sure as this is quite expensive, and random noise may also work (see below), but this brings a level of understanding and analytical consistency that makes the study worth the trouble. I especially like theorem 1, and how it shows that adversarial training should provable tame the Hessian sharpness (though it is not well placed in context) The reviewer has found that very large batch sizes is critical for performance on noisy data (a similar results was just published in https://arxiv.org/pdf/1705.10694.pdf), which suggests that the noise could also be random rather than adversarial, and that the 2 strategy should be compared. I failed to understand figure 5: - why is the x-axis varying beyond[0,1] if it represents lambda - what data is the cross-entropy computed over: non-adversarial as the original model seems to have a lower loss - what does the curvature mean? - how can you average over deep models? It usually does not work unless on is just a small perturbation of the other?

Reviewer 3



Summary: This paper consists of two parts. The first part claims that using gradient descent to train deep network with larger batch size, the parameter \theta converges to areas which have larger Hessian spectrum. The second part claims that with larger batch size, the network model is less robust. It also provides a side result that finding a perturbation direction is essentially a saddle-free problem, which makes this problem easier. Although there exists work which proposed methods to make good use of big batch [1] [2], this paper still concludes that big batch size have negative impact on training, based on several experiments. Strengths: This work designs several experiments, including visualizing the landscape of the loss, comparing the spectrum between robust model and original model, to backup its main claims. The experiment results seem reliable. These experiments almost cover every aspects that people need to show about the effect of batch size and the relationship between the batch size and the robustness. Meanwhile, the way the authors interpret the experimental results seems reasonable. The results delivered in this paper will have great impact on both the theory and practice of the deep learning. Weaknesses: The major weakness of this work is that it only runs experiments on two small datasets MNIST and CIFAR10. Since modern deep learning training task is much more complex, and the behavior of the network model heavily depends on the sample size $n$, the conclusion the authors draw will be more convincing if the authors can run experiments on datasets with at least moderate size, where $10^5 < n < 10^8$. [1] Priya Goyal, Piotr Doll´ar, Ross Girshick, Pieter Noordhuis, Lukasz Wesolowski, Aapo Kyrola, Andrew Tulloch, Yangqing Jia, and Kaiming He. “Accurate, large minibatch SGD: training imagenet in 1 hour”. In: arXiv preprint arXiv:1706.02677 (2017). [2] Samuel L Smith, Pieter-Jan Kindermans, and Quoc V Le. “Don’t decay the learning rate, increase the batch size”. In: arXiv preprint arXiv:1711.00489 (2017). ======== Thanks for the authors' response.